# A Label-Free Electrochemical Biosensor for Homocysteine Detection Using Molecularly Imprinted Polymer and Nanocomposite-Modified Electrodes

**DOI:** 10.3390/polym15102241

**Published:** 2023-05-09

**Authors:** Unchalee Kongintr, Benchaporn Lertanantawong, Chamras Promptmas

**Affiliations:** 1Biosensor Laboratory, Department of Biomedical Engineering, Faculty of Engineering, Mahidol University, Nakhon Pathom 73170, Thailand; unchalee.ko@gmail.com (U.K.); benchaporn.ler@mahidol.ac.th (B.L.); 2Faculty of Medical Technology, Huachiew Chalermprakiat University, Samut Prakan 10540, Thailand

**Keywords:** molecularly imprinted polymer, carbon nanotube, nanocomposite, biosensor, electrochemical biosensor, homocysteine

## Abstract

An essential biomarker for the early detection of cardiovascular diseases is serum homocysteine (Hcy). In this study, a molecularly imprinted polymer (MIP) and nanocomposite were used to create a label-free electrochemical biosensor for reliable Hcy detection. A novel Hcy-specific MIP (Hcy-MIP) was synthesized using methacrylic acid (MAA) in the presence of trimethylolpropane trimethacrylate (TRIM). The Hcy-MIP biosensor was fabricated by overlaying the mixture of Hcy-MIP and the carbon nanotube/chitosan/ionic liquid compound (CNT/CS/IL) nanocomposite on the surface of a screen-printed carbon electrode (SPCE). It showed high sensitivity, with a linear response of 5.0 to 150 µM (R^2^ of 0.9753) and with a limit of detection (LOD) at 1.2 µM. It demonstrated low cross-reactivity with ascorbic acid, cysteine, and methionine. Recoveries of 91.10–95.83% were achieved when the Hcy-MIP biosensor was used for Hcy at 50–150 µM concentrations. The repeatability and reproducibility of the biosensor at the Hcy concentrations of 5.0 and 150 µM were very good, with coefficients of variation at 2.27–3.50% and 3.42–4.22%, respectively. This novel biosensor offers a new and effective method for Hcy assay compared with the chemiluminescent microparticle immunoassay at the correlation coefficient (R^2^) of 0.9946.

## 1. Introduction

Cardiovascular disease diagnosis is gaining great attention globally because of its prevalence and the high mortality rate. Methods for the detection of early inflammation in cardiovascular disease include screening methods such as blood pressure tests, electrocardiography (ECG), exercise stress tests, and CT scans [1]. Increased levels of cardiovascular disease markers in human serum, which are creatinine kinase MB subform (CK-MB), Troponin I (cTnI), Troponin T (cTnT), Myoglobin, C-reactive protein (CRP), Myeloperoxidase (MPO), heart fatty acid binding protein (H-FABP), and homocysteine (Hcy), are a reliable symptom associated with cardiovascular disease patients. The determination of inflammation via cardiovascular markers plays an important role in the early diagnosis of cardiovascular disease [2,3,4]. In clinical laboratory assays, the detection methods for cardiovascular markers include high-performance liquid chromatography [5,6,7,8,9,10], gas chromatography–mass spectrometry (GC-MS) [11,12], capillary electrophoresis [13,14], and photoluminescence assays [15]. All these methods require laborious derivatization processing and sophisticated instruments. This is a time-consuming process. The operation costs of specific instruments and reagents also limit their wider application in clinical laboratories. Hence, the assay method based on the immunological reaction was developed, with its low operation costs and suitability for point-of-care testing [16,17,18,19].

Recently, the electrochemical method for Hcy detection was extensively developed using enzyme-based electrodes [20,21] and screen-printed electrodes (SPE) [22]. The screen-printed electrode is versatile, simple, low-cost, easily operated, small-sized, portable, and capable of mass production. To improve the electrochemical signal, nanomaterials were applied to SPE, such as gold nanoparticles [23,24,25], carbon nanotubes [26,27,28], graphene [29,30], and nanocomposite materials [31]. The sensitivity of Hcy detection was noticeably improved by applying quantum dots to the assay platform [15,32,33].

To date, the aptamers and the molecularly imprinted polymers (MIPs) have attracted substantial interest as a biomimetic recognition element to solve the limitations of antibodies. They exhibit interesting selectivity and high affinity in binding to specific targets such as antibodies, but demonstrate more stability and reusability than antibodies [34,35,36,37,38]. The aptamer has been established for Hcy detection using voltammetric assays and optical detection methods [39,40,41,42]. They showed superior signal detection performance, specificity, reproducibility, and acceptable accuracy when working with real serum samples [39,40,41,42].

Generally, MIPs are synthesized by the polymerization of functional monomers and cross-linker monomers, which have prior interactions with target molecules, as a template. After polymerization, the template is subsequently removed from the complex matrix to open cavities with a size and shape complementary to the template. MIPs function as artificial specific receptors to target molecules with a binding function, such as antibodies [43]. To date, MIPs have been reported as selective recognition elements for a range of both biological and chemical molecules, including amino acids and proteins [44,45,46] and drugs [47], because of their robustness and tolerance to ambient temperatures.

MIPs for Hcy have been fabricated and applied for Hcy detection by in situ fluorescent derivatizations in the optical sensor system. None of the analytical characteristics have been presented due to the sensitivity of detection, which needs to be improved in practical applications [45]. Recently, the use of electrochemical methods incorporating MIPs has emerged as a well-established analytical technique employing the concept of the selective uptake of an analyte of interest and subsequent generation of a characteristic electrochemical signal.

In this research, a label-free electrochemical biosensor system for the detection of serum Hcy was created by the surface modification of screen-printed carbon electrodes (SPCE) with the combination of synthesized MIPs and a nanocomposite (carbon nanotube/chitosan/ionic liquid). MIP-modified carbon paste electrodes were fabricated, characterized, and applied for the electrochemical detection of Hcy by differential pulse voltammetry. This novel detection system provides a promising solution for the preliminary detection of hyperhomocysteinemia and other diseases associated with homocysteine.

## 2. Materials and Methods

### 2.1. Chemicals and Specimens

Homocysteine (Hcy), methacrylic acid (MAA), 2,2′-azobis(2-methylpropio nitrile) (AIBN), trimethylolpropane trimethacrylate (TRIM), methionine, cysteine, L-ascorbic acid, multi-walled carbon nanotubes (MWCNT-COOH) (outer diameter x length 7–15 nm x 0.5–10 µm), 1-butyl-3-methylimidazolium tetrafluoroborate (BF4) and potassium chloride (KCl), methanol, acetic acid, potassium hexacyanoferrate (K_4_[Fe(CN)_6_]), and chitosan (CS) were obtained from Merck KGaA, Darmstadt, Germany. All chemicals were of analytical grade and all solutions were prepared with ultrapure water (resistivity ≥ 18 MΩ cm) from a Millipore Milli Q system obtained from Merck KGaA, Darmstadt, Germany. All the solutions for electrochemical analysis were dissolved in 0.1 M phosphate buffer solution (PBS), pH 7.

Eighteen anonymous serum samples were left-over samples from the Clinical Chemistry Laboratory Service Unit of the Faculty of Medical Technology, Huachiew Chalermprakiat University.

### 2.2. Apparatus

A Thermo Scientific Nicolet iS5 FTIR Spectrometer from Thermo Fisher Scientific Inc. (Waltham, MA, USA) was used to record the Fourier transform infrared (FTIR) spectra. An ER466, eDAQ electrochemical workstation from eDAQ Pty Ltd. (Sydney, NSW, Australia), coupled with a three-electrode cell format, was used for cyclic voltammetric (CV) and differential pulse voltammetric (DPV) studies. Triple screen-printed carbon electrodes (SPCE) of 3 mm diameter carbon working electrodes, a solid-state silver/silver chloride (Ag/AgCl) reference electrode, and carbon counter-electrodes were obtained from Quasense Co., Ltd., Bangkok, Thailand.

### 2.3. Procedures

This platform uses molecularly imprinted polymer and nanocomposite-modified working carbon paste electrodes. The SPCE was fabricated with the mixture of Hcy-MIP and the CNT/CS/IL nanocomposite to obtain a Hcy-MIP biosensor electrode, as shown in Figure 1a. The Hcy in the serum sample binds with Hcy-MIP on the surface of the electrode. As illustrated in Figure 1b, this binding event can be identified by measuring the electrochemical signal of differential pulse voltammetry (DPV) employing K_4_[Fe(CN)_6_]^3−/4−^.

#### 2.3.1. Preparation of Molecularly Imprinted Polymers

According to Chow et al. [45], the molecularly imprinted polymer (MIP) was created via precipitation polymerization with Hcy as the target molecule. The mixture solution containing 1 mM Hcy as a molecular template and 2 mM MAA as a functional monomer in 50 mL of methanol was agitated for 12 h. The monomer mixture was then mixed with 10 mM TRIM as a cross-linking agent and 1 mM AIBN as a radical-induced polymerization initiator. The mixture was agitated for 15 min under a nitrogen atmosphere and maintained at 60 °C for 24 h to complete polymerization. The polymerized product of Hcy containing MIP was decanted and washed with methanol. The Hcy template was eliminated by refluxing with 100 mL of a 10% NaOH solution for 1 h, followed by cleaning with a 1% acetic acid solution and deionized water (DI) until the product suspension reached a pH of 7.0. The Hcy-MIP precipitate was filtered and dried with acetone. A non-imprinted polymer (NIP) was created by using the Hcy-MIP preparation procedure without exposure of Hcy.

#### 2.3.2. Preparation of CNT/CS/IL Nanocomposite

The nanocomposite for the Hcy-MIP biosensor was prepared according to Gopalan et al. [48] by dispersing 1 mg of MWCNT in 1 mL DI water with sonication for 2 h. The solution of 0.1% chitosan (CS) in acetic acid was dropped into the CNT suspension and sonicated for 1 h. To form a complete CNT/CS/IL nanocomposite, 10 µL of BF4 was added and it was sonicated for 30 min.

#### 2.3.3. Hcy-MIP Electrode Fabrication

The Hcy-MIP biosensor was fabricated by the modification of SPCE with the mixture of Hcy-MIP and the CNT/CS/IL nanocomposite. The Hcy-MIP electrode was prepared by overlaying the surface of SPCE with 1 µL of a CNT/CS/IL nanocomposite suspension containing Hcy-MIP at the optimal concentration for 1 h at room temperature.

After being air dried, this Hcy-MIP-modified electrode was kept at 4 °C. An identical process to that used to generate the Hcy-MIP-modified electrode was used to prepare the NIP-modified electrode, but only NIP was used in place of Hcy-MIP. The optimal response to Hcy was examined for the concentrations of Hcy-MIP of 5, 25, 50, 75, 100, and 150 µM. The Hcy-MIP- and NIP-modified electrodes were characterized by a Nicolet 6700 FTIR spectrometer to demonstrate the chemical bonding on the tested surface. The attenuated total reflectance (ATR) mode IR spectrum was recorded in the range of 4000 to 625 cm^−1^ at a resolution of 2 cm^−1^ with 64 scans.

#### 2.3.4. Characterization and Electroanalytical Measurements of the Hcy-MIP Electrodes

The Hcy-MIP-modified electrodes were characterized by cyclic voltammetry (CV) and differential pulse voltammetry (DPV) using K_4_[Fe(CN)_6_]^3−/4−^ as an electroactive probe molecule. The electrochemical behavior of Hcy-MIP/CNT/CS/IL/SPCE in 10 mM ferro-ferricyanide (K_4_[Fe(CN)_6_]) and 1 M KCl in 0.1 M PBS pH 7.0 was investigated and compared with that of non-modified and NIP modified electrodes. The electrochemical signals of the non-modified, Hcy-MIP-, and NIP-modified electrodes were investigated in the same solution unless mentioned otherwise. In this investigation, CV analyses were performed at a scan rate of 10 mV/s in the potential range of −0.5 to 0.8 V. DPVs were conducted in the same potential range with an amplitude of 0.05 V, pulse width of 0.05 s, and a pulse period of 0.5 s. Prior to recording the response, the electrode was given 10 min to interact with the analyte.

#### 2.3.5. Determination of Homocysteine

The studies were conducted in their entirety at room temperature. The reaction of the Hcy-MIP biosensor started by applying 150 µL of Hcy standard solution or tested sample and allowing its complete interaction with the active sites on the Hcy-MIP biosensor surface for 10 min. The electrode was then carefully cleansed with deionized water to remove any potential non-specific binding substances. The DPV peak current of K_4_[Fe(CN)_6_ was observed from the reaction using the DPV and scanning the electrode potential from −0.5 to 0.8 V. DPVs were recorded for the Hcy-MIP biosensor.

The optimal conditions for the Hcy-MIP biosensor were investigated, namely the appropriate incubation time, pH, and Hcy concentration for an effective Hcy measurement. The Hcy-MIP biosensor was connected to an ER466, eDAQ electrochemical workstation as the DPV signal detection system. To accomplish the function of the Hcy-MIP biosensor, 150 µL of 100 µM Hcy in 0.1 M PBS, pH 7.0, with 1 M KCl was applied to the Hcy-MIP biosensor and incubated at different times from 1 to 30 min; it was then washed out to avoid any possible non-specific interaction by rinsing with DI water. The peak current of K_4_[Fe(CN)_6_ was observed from the reaction using the DPV by scanning the electrode potential from −0.5 to 0.8 V to obtain an appropriate incubation time for Hcy determination. To obtain the optimal pH and Hcy concentrations in the reaction, 0.1 M PBS at a pH of 7.0, 7.2, 7.4, 7.8, and 8.0, and Hcy solutions at concentrations of 5, 25, 50, 75, 100, and 150 µM, were applied to the Hcy-MIP biosensor for 10 min. The current signals of K_4_[Fe(CN)_6_]^3−/4−^ were then recorded by scanning the electrode potential from −0.5 to 0.8 V after rinsing with DI water.

The optimal reaction time, reaction pH, and dose response for Hcy detection were obtained by plotting the peak current from the DPV against the incubation time, the pH of the buffer, and the Hcy concentrations, respectively.

#### 2.3.6. Evaluation of Hcy-MIP Biosensor Performance

The analytical precision of Hcy detection by the Hcy-MIP biosensor was determined via the intra- and inter-assay variation by measuring Hcy at the concentration of 5 and 150 µM. Both the intra- and inter-assay variation were obtained via 20 measurements and each measurement was conducted in triplicate under optimal conditions. The mean of the replicates and the standard deviation (SD) were used to calculate the coefficient of variation (CV) of this evaluation.

The limit of detection (LOD) of Hcy was determined using the 3-standard-deviation (3SD) values of the test results of 1 M Hcy in PBS, pH 7.0, under optimal conditions with 10 repetitions.

The analytical accuracy of the Hcy-MIP biosensor was determined via a recovery assay. A spike recovery was employed by adding known concentrations of 50, 75, 100, and 150 µM Hcy. The recovery as a percentage was calculated as the ratio of the observed Hcy concentration to that of the spiked concentration.

The specificity of the assay was evaluated by exposing the Hcy-MIP biosensor to 500 µM of several physiological substances: cysteine, methionine, glutathione, and ascorbic acid. The analysis was performed under the same procedure as the Hcy assay.

Additionally, the performance of the Hcy-MIP biosensor was evaluated by determining the Hcy concentration in the serum sample. The Hcy-MIP biosensor measurement and the chemiluminescent microparticle immunoassay (CMIA) method often used in clinical laboratories were compared using 18 serum samples. The correlation between these two measuring techniques was assessed using paired-sample t-test analysis and correlation analysis.

## 3. Results and Discussion

### 3.1. Characterization of Hcy-MIP

The morphology of the imprinted polymer for homocysteine was observed under scanning electron microscopy (SEM). The SEM images in Figure 2 show the spherical shape of the Hcy-MIP and NIP particles with the approximate diameters of 100 ± 10 nm and 150 ± 15 nm, respectively. These were consistent with most of the imprinted polymer particles prepared by the precipitation method [45]. The chemical characterization or elemental analysis of these imprinted particles was also observed by the FTIR spectrometer. The FTIR spectra in Figure 3 present the unique chemical characteristics of the polymerized compounds of MAA both in Hcy-MIP and NIP particles (red dashed line). The peaks at 2920 cm^−1^ and 3438 cm^−1^ indicate the stretching vibrations of the methyl C-H and O-H groups, respectively. The peak at 1720 cm^−1^ represents the >C=O, 1631 cm^−1^ the alkenyl >C=C< stretching, 1463 cm^−1^ and 1386 cm^−1^ the methyl C-H bending, 1295 cm^−1^ the vinylidene bending, and 1140 cm^−1^ the C-O stretching. Hcy (green solid line) had its characteristic peak at 2910 cm^−1^ due to C-H stretching from tertiary carbon, at 2080 cm^−1^ due to C-O stretching, 1520 cm^−1^ due to N-H bending, 1322 cm^−1^ due to >C-C< stretching,1060 cm^−1^ due to C-N stretching, 853 cm^−1^ due to S-H bending, 758 cm^−1^ due to methylene CH2 rocking, and 691 cm^−1^ due to –CH2-SH bending.

The IR spectrum showed decreasing signals of C=O and C-O and an increase in the C-N peak at 1060 cm^−1^ and N-H at 1520 cm^−1^ when Hcy bound to Hcy-MIP (blue line). These changes were due to the bonding of the H atom of Hcy and O atom of MIP.

### 3.2. Electrochemical Characteristics of Hcy-MIP

The CV analysis of all electrodes was conducted using K_4_[Fe(CN)_6_]^3−/4−^ as an electrochemical marker. Figure 4 shows the CV of a different modified electrode from the nanocomposite-modified electrode (CNT/CS/IL/SPCE), MIP/CNT/CS/IL/SPCE, MIP/CNT/CS/IL/SPCE after interaction with 1 mM Hcy for 10 min, and NIP/CNT/CS/IL/SPCE. The cyclic voltammetric current response of the nanocomposite-modified electrode was found to decrease with the incorporation of non-conducting polymer material as a modification, but the porous nature of the MIP produced due to the leaching of Hcy made it more responsive than the NIP. The current response further decreased after 10 min of MIP/CNT/CS/IL/SPCE exposure to 1 mM Hcy because the Hcy molecules formed hydrogen bonds with the MIP on the electrode surface, decreasing the surface area available for electron transfer. The unavailability of the pores in NIP resulted in a very poor current response for the corresponding electrode.

CVs of MIP/CNT/CS/IL/SPCE were also recorded at various scan rates of 10, 20, 30, 40, 50, 60, 70, 80, 90, and 100 mV/s in the same potential range and are shown in Figure 5a–j (inset: Ip against scan rate). A surface-controlled electrochemical reaction system was found since the peak current (Ip) was seen to grow linearly with the scan rate. Thus, the ferro–ferri conversion redox reaction for the MIP-modified electrode appeared to be an absorption-controlled process; initially, it was absorbed onto the electrode surface and subsequently underwent a redox reaction.

### 3.3. Optimization of Hcy-MIP Biosensor

#### 3.3.1. Concentration of MIP in Hcy-MIP Biosensor

Different MIP-modified SPCEs were prepared with MIP at the concentrations of 0.1, 0.2, 0.5, 0.7, and 1.0 mg/mL. The current response from DPV was observed after interaction with 0.1 mM Hcy for 10 min. The electrode response in Figure 6 demonstrates the positive relationship between the current signal and MIP at a concentration up to 0.5 mg/mL. Using MIP greater than 0.5 mg/mL did not significantly increase the current signal. As a result, 0.5 mg/mL MIP is appropriate for the preparation of the Hcy-MIP biosensor.

#### 3.3.2. Effect of pH

The DPV response of the Hcy-MIP biosensor was recorded after 0.1 mM Hcy was applied for 10 min, followed by 10 mM ferro–ferricyanide solution in 0.1 M PBS at various pHs of 7.0, 7.2, 7.4, 7.8, and 8.0. The DPV was used to monitor the pH while the biosensor function was at its peak, as shown in Figure 7. The electrode response was seen to increase initially when the solution’s pH rose to a maximum, and then it decreased. Since this biosensor responded most strongly to Hcy at pH 7.0, it was decided that this was the optimal pH for Hcy determination.

### 3.4. Determination of Hcy

The DPV response after allowing the interaction between the Hcy-MIP biosensor and Hcy for 10 min served to exhibit the performance of the Hcy-MIP biosensor at various Hcy concentrations of 0, 25, 50, 75, 100, and 150 µM, as shown in Figure 8. The corresponding calibration curve in Figure 8 was obtained, with an R^2^ correlation coefficient of 0.9753. The limit of detection (LOD) of the Hcy-MIP biosensor was 1.2 µM (S/N = 3). The DPV response of MIP/CPE for the K_4_[Fe(CN)_6_]^3−/4−^ redox couple was found to decrease with an increase in the Hcy concentration. The decrease in the availability of the pores at the electrode surface was caused by a rise in the Hcy concentration because the interconversion of the Fe(II)–Fe(III) redox reaction unfolds as a surface-controlled process. Additionally, the response current decreased as a result of the imprinted sites being blocked by the hydrogen-bonded Hcy, which also reduced the effective surface area.

### 3.5. Evaluation of Hcy-MIP Biosensor Performance

#### 3.5.1. Dose Response of Hcy-MIP Biosensor

The dose response curve for the determination of the Hcy-MIP biosensor is represented in Figure 9. This curve was plotted between the current signal obtained from the DPV of potassium ferrocyanide and the Hcy concentration in the range of 5.0 to 150 µM. The DPV peak current inversely correlated to the Hcy concentration was 19.40 ± 0.145, 19.07 ± 0.231, 17.81 ± 0.172, 17.56 ± 0.211, 16.91 ± 0.218, and 15.78 ± 0.371 µA, respectively. The sensitivity for Hcy of this Hcy-MIP biosensor was 25.2 nA/µM. This Hcy-MIP biosensor had a linear detection range between 5.0 and 150 µM and a high correlation coefficient (R^2^ = 0.9753).

Additionally, the limit of detection (LOD) of the Hcy-MIP biosensor was defined as three times the standard deviation of the assay results of 1 µM Hcy in PBS, pH 7.0, under optimal conditions in 10 replicates. It was 1.2 µM, which is comparable to those calculated from other reported methods, such as an enzyme-based biosensor platform [20], an aptamer-modified gold nanoparticle/graphene sponge electrode platform [39], and a rapid liquid chromatography–tandem mass spectrometry platform [12], as shown in Table 1. However, some methods have LODs that are significantly lower than this Hcy-MIP biosensor method—for instance, the quantum dot platform method with an LOD and detection range in the nM range [15,32,33] or HPLC with a post-column reaction platform, where the LOD and detection range are 10 times better [5,8,9]. It is possible to modify the Hcy-MIP to obtain a fluorescent signal [45] and apply it to the fluorescent sensor platform to obtain a better LOD [49]. During the polymerization process, the Hcy-MIP can be coated on a polymeric photonic platform for the real-time sensing of Hcy [50]. The normal range of serum Hcy is typically 5–16 µM. The range of hyperhomocysteinemia can also be classified as mild (16–30 µM), moderate (30–100 µM), and severe (>100 µM). The linear response range of this biosensor is 5.0 to 150 µM with an LOD at 1.2 µM, which covers both pathological and normal plasma levels of homocysteine without the requirement for sample dilution.

#### 3.5.2. Analytical Accuracy

A recovery assay was employed to assess the analytical accuracy of the Hcy-MIP biosensor. Four Hcy concentration levels of 50, 75, 100, and 150 µM were added and examined for the recovery of Hcy. According to Table 2, the recovered spike Hcy samples had concentrations of 46.1, 72.6, 94.4, and 144.3 µM or 91.10%, 96.07%, 93.85%, and 95.83% recovery, respectively, with an average recovery percentage of 94.21.

#### 3.5.3. Analytical Precision

Three Hcy concentration levels of 5 and 150 µM were chosen for the analytical precision of the Hcy-MIP biosensor for both the intra- and inter-assay methods. Twenty measurements were taken for the intra-assay in the same batch, whereas twenty measurements were taken for the inter-assay over the course of twenty days under optimal conditions. Calculations were made to convert the DPV peak current signal to the Hcy concentration. The coefficients of variation (CVs) for the intra- and inter-assay at the concentrations of 5 and 150 µM were 2.27%, 3.50%, and 3.42%, 4.22%, respectively, as shown in Table 3. According to this finding, the Hcy-MIP biosensor might be used to determine Hcy with satisfactory reproducibility.

#### 3.5.4. Analytical Specificity

The specificity of the Hcy-MIP biosensor was examined by recording the DPVs before and after exposure to 0.5 mM of cysteine, methionine, glutathione, and ascorbic acid. The insignificant current response observed for the interferents demonstrates the considerable selectivity of this biosensor, which is attributed to the Hcy-specific cavities present in the imprinted polymer matrix. This demonstrates that the selectivity of this Hcy-MIP biosensor is suitable for diagnostic application.

#### 3.5.5. Comparative Assay

Eighteen serum samples were selected for the comparative assay between the Hcy-MIP biosensor and chemiluminescent microparticle immunoassay method (CMIA). The statistical analysis of 18 measurements from the MIP sensor and CMIA was performed using a paired t-test. The correlation value of the two measurement methods was 0.9946, with a non-significantly different measurement result (paired t-test: *p* > 0.01), as shown in Figure 10. Therefore, this proposed Hcy-MIP biosensor could be applied as a testing method to determine serum Hcy concentrations at both normal and abnormal levels.

## 4. Conclusions

An MIP of Hcy was synthesized from MAA, characterized by FTIR, and was used in the modification of SPCE, which was further characterized with CV and DPV. CV and DPV analysis showed that the electrode was responsive to the target molecule. The MIP/modified SPCE as the Hcy-MIP biosensor was optimized at 0.5 mg/mL, which provided the highest electrode DPV response. This innovative biosensor combines synthesized MIPs with a nanocomposite made of carbon nanotubes, chitosan, and ionic liquid, which has the advantages of a label-free electrochemical system and extreme stability of the imprinted polymer on the electrode surface. The detection of Hcy by the Hcy-MIP biosensor was determined by the DPV peak current in the range of 5.0 to 150 µM, with a limit of detection of 1.2 µM and correlation coefficient, R^2^, of 0.9753. The developed biosensor was only selective to Hcy as it produced no electrode response towards physiological interferences such as methionine, cysteine, and L-ascorbic acid. Thus, the proposed biosensor exhibits benefits including ease of electrode construction, excellent electrode integrity, and a discernible detection limit and detection range.

## Figures and Tables

**Figure 1 polymers-15-02241-f001:**
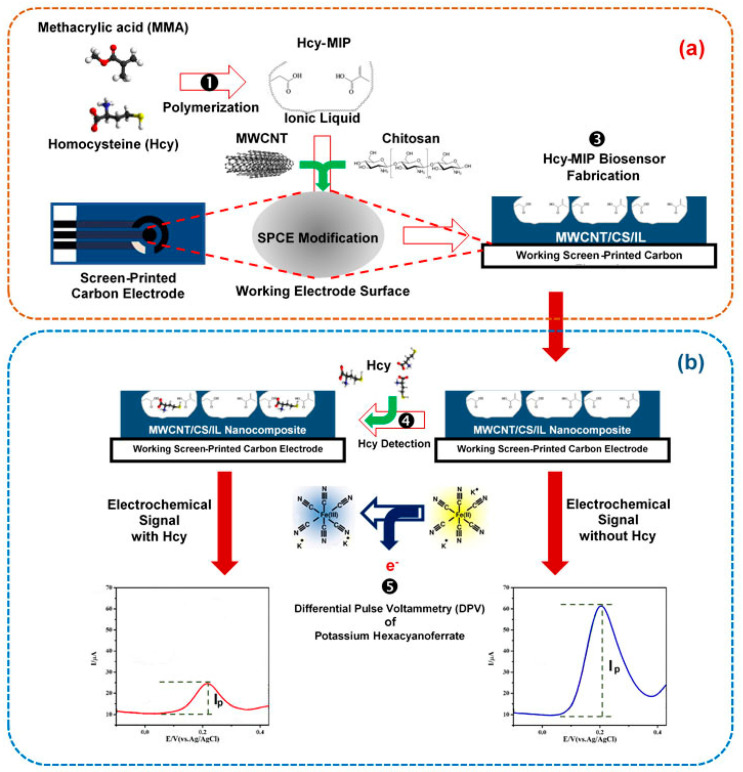
Scheme for homocysteine determination with the Hcy-MIP electrochemical biosensor platform. (**a**) The SPCE was fabricated with the mixture of Hcy-MIP and CNT/CS/IL nanocomposite to obtain Hcy-MIP biosensor electrode; (**b**) the detection was performed by measuring the electrochemical signal of differential pulse voltammetry (DPV) using K_4_[Fe(CN)_6_]^3−/4−^ as an electroactive probe molecule.

**Figure 2 polymers-15-02241-f002:**
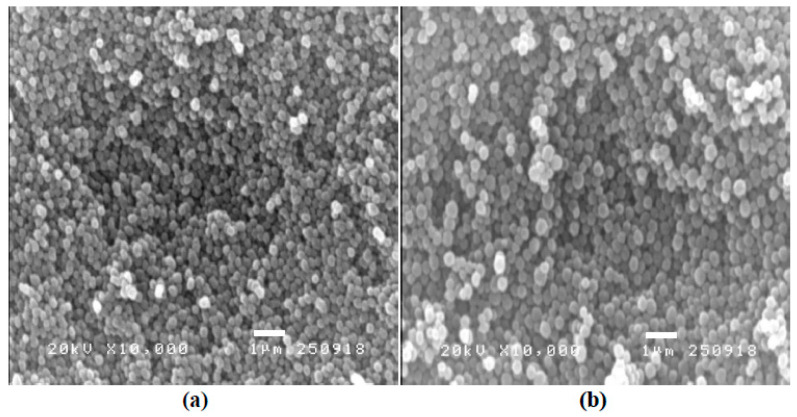
The SEM images of the imprinted polymer: (**a**) homocysteine-imprinted polymer (Hcy-MIP), (**b**) non-imprinted polymer (NIP).

**Figure 3 polymers-15-02241-f003:**
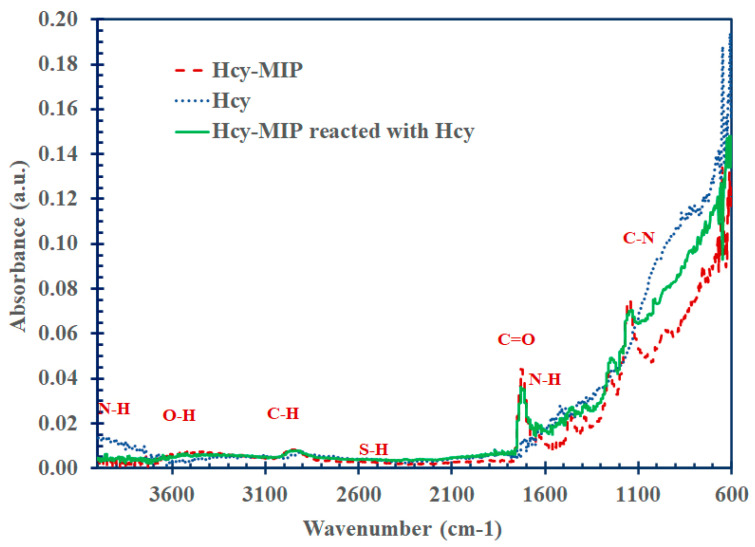
The FTIR spectrophotometry results of homocysteine-imprinted polymer (Hcy-MIP) (red dashed line), homocysteine (blue dotted line), and Hcy-MIP reacted with Hcy (green solid line).

**Figure 4 polymers-15-02241-f004:**
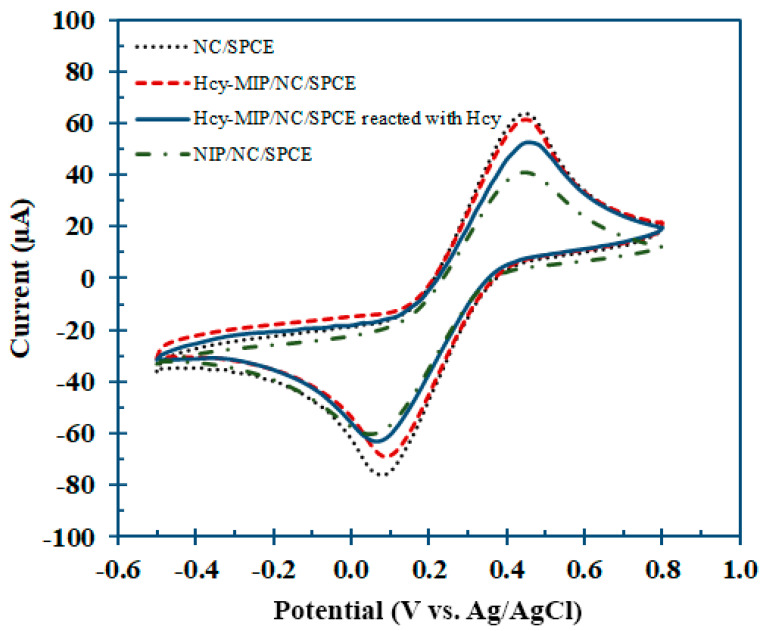
The CV analysis of modified electrodes with nanocomposite-modified SPCE (CNT/CS/IL/SPCE) (dotted line), Hcy-MIP/CNT/CS/IL/SPCE (dashed line), Hcy-MIP/CNT/CS/IL/SPCE after interaction with 1 mM Hcy for 10 min (solid line), and NIP/CNT/CS/IL/SPCE (dashed-dotted line). The electrochemical measurement was conducted in 10 mM K_3_[Fe(CN)^6^] in 0.1 M phosphate buffer containing 1 M KCl.

**Figure 5 polymers-15-02241-f005:**
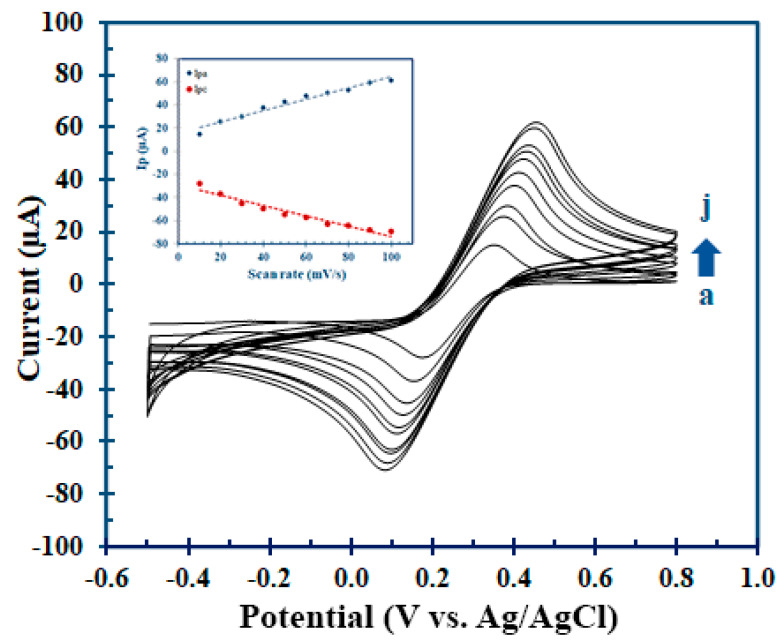
The CVs of MIP/CNT/CS/IL/SPCE at different scan rates of (a) 10, (b) 20, (c) 30, (d) 40, (e) 50, (f) 60, (g) 70, (h) 80, (i) 90, and (j) 100 mV/s in the potential range of −0.5–0.8 V (vs./Ag/AgCl) (inset: Ip vs. scan rate).

**Figure 6 polymers-15-02241-f006:**
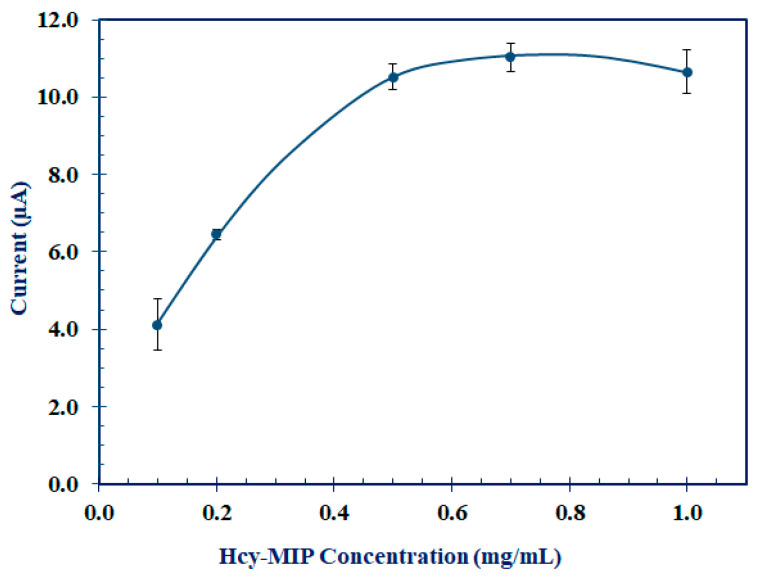
The DPV peak current response of MIP/CNT/CS/IL/SPCE (Hcy-MIP biosensor) fabricated with Hcy-MIP at the concentration of 0.1, 0.2, 0.5, 0.7, and 1.0 mg/mL.

**Figure 7 polymers-15-02241-f007:**
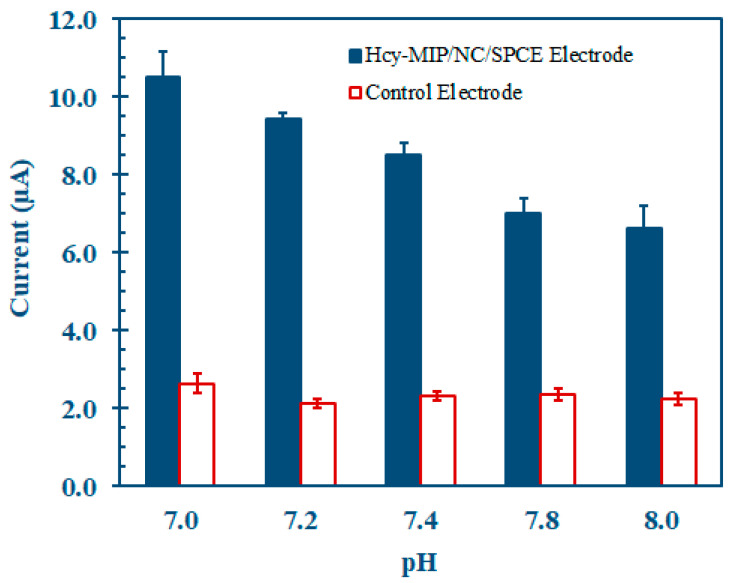
The effect of pH on the DPV peak current response of MIP/CNT/CS/IL/SPCE (Hcy-MIP biosensor), after interaction with 0.1 mM Hcy for 10 min. The electrochemical measurement was performed in 10 mM ferro–ferricyanide in 0.1 M PBS at different pHs of 7.0, 7.2, 7.4, 7.8, and 8.0 solution containing 1 M KCl.

**Figure 8 polymers-15-02241-f008:**
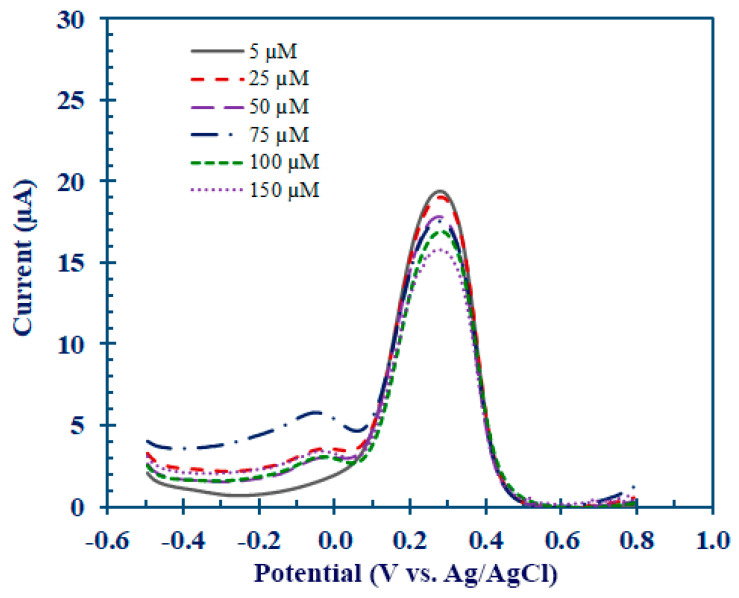
The DPV signal response of Hcy-MIP biosensor after interacting with Hcy at concentrations of 5, 25, 50, 75, 100, and 150 µM for 10 min. The electrochemical measurement was conducted in 10 mM K_3_[Fe(CN)_6_] in 0.1 M phosphate buffer pH 7.0 containing 1 M KCl.

**Figure 9 polymers-15-02241-f009:**
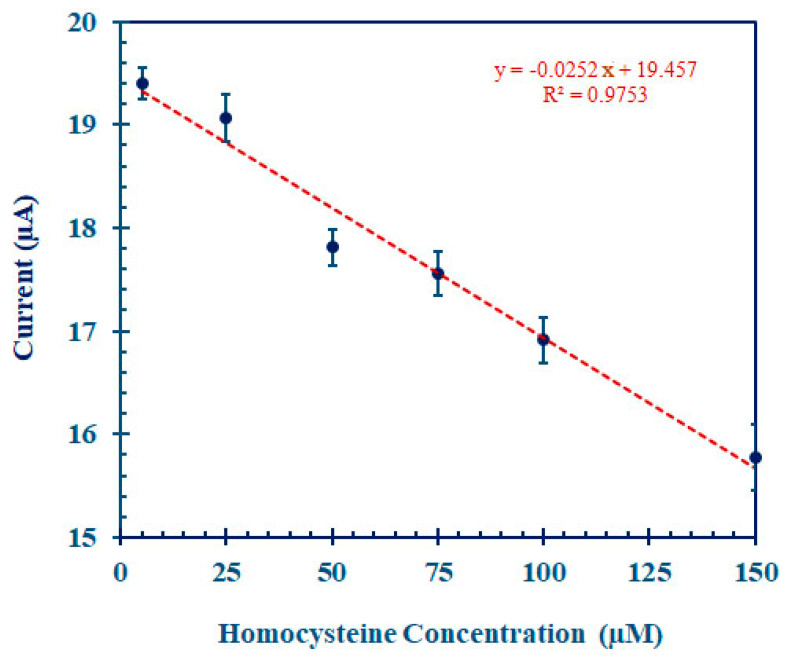
Dose response curve for homocysteine determination. The Hcy-MIP biosensor reacted with a standard solution containing 5–150 µM Hcy for 10 min at room temperature. The DPV peak current response was then measured by applying 10 mM K_3_[Fe(CN)_6_] in 0.1 M phosphate buffer, pH 7.0, containing 1 M KCl. Values are mean ± S.D. (*n* = 3).

**Figure 10 polymers-15-02241-f010:**
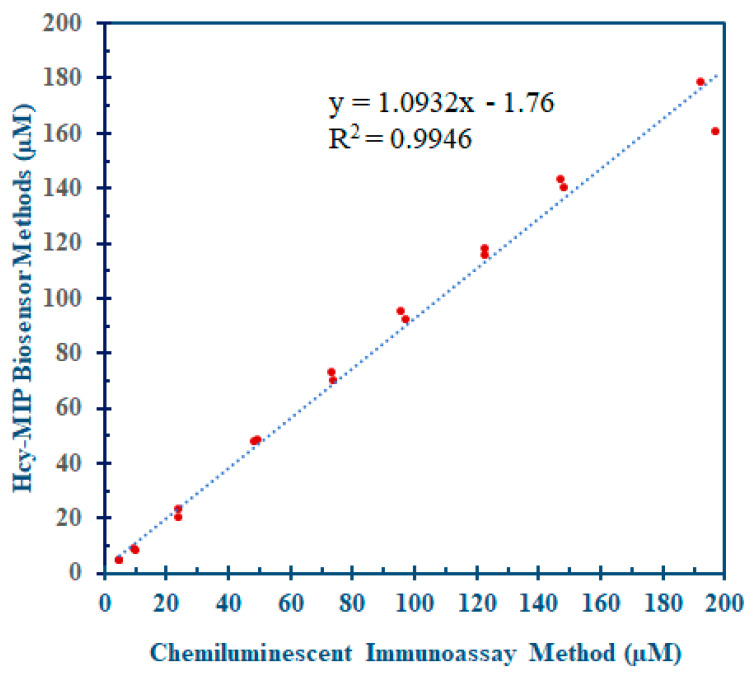
The comparative assay of serum homocysteine between Hcy-MIP biosensor and chemiluminescent microparticle immunoassay method (CMIA) with a correlation value of R^2^ = 0.9946.

**Table 1 polymers-15-02241-t001:** Comparison of limit of detection (LOD) of other previously reported homocysteine detection methods.

Platform	Methods	LOD	Linearity	Ref.
**Chromatographic Platform**				
Simultaneous liquid chromatography–mass spectrometry	HPLC-MS	0.75 µM	1.5–740 µM	[11]
Rapid liquid chromatography–tandem mass spectrometry	HPLC-MS/MS	1.0 µM	0.0–61.6 µM	[12]
HPLC with electrochemical coulometric array detection	HPLC-ED	0.14 µM	-	[6]
HPLC with platinum/poly(methyl violet) (Pt/MV)-modified electrode	HPLC-ED	0.1 µM	0.2–100 µM	[7]
Thiocarbonyldiimidazole (TCDI) post-column reaction	HPLC-UV	0.1 µM	2.5–10 µM	[8]
2-chloro-1-methylpyridinium iodide (CMPI) post-column reaction	HPLC-UV	0.1 µM	0.5–50 µM	[9]
Methanolic monobromobimane for thiol derivatization	HPLC-FL	0.12 µM	3.9–62 µM	[5]
Iodoacetylaminobenzanthrone (IAB) post-column reaction	HPLC-FL	2.3 nM	0.05–25 µM	[10]
**Electrophoresis Platform**				
Capillary electrophoresis/electrochemistry	Amperometry	0.5 µM	1–100 µM	[13]
Capillary electrophoresis with pyrroloquinoline quinone-modified electrode	Amperometry	0.03 µM	0.1–5 µM	[14]
**Immunoassay Platform**				
Lateral flow immunofluorescent	Optical	0.27 µM	1.0–50 µM	[16]
**Enzyme-Based Biosensor Platform**				
Amino acid oxidase immobilized on screen-printed carbon electrode	Amperometry	NA	6.4–100 µM	[22]
Amino acid oxidase immobilized on oxygen electrode	Potentiometry	2.0 µM	0.05–1.5 µM	[20]
Homocysteine desulfhydrase enzyme electrode	Potentiometry	NA	0.15–1.8 µM	[21]
**Nanomaterial-Based Biosensor Platform**				
Cytochrome c-anchored gold nanoparticles on screen-printed electrode	Amperometry	0.3 µM	0.4–700 µM	[23]
Gold nanoparticle-incorporated reduced graphene oxide electrode	Amperometry	6.9 µM	2–14 µM	[24]
Reduced graphene oxide–TiO_2_ (RGO-TiO_2_) nanocomposite on glassy carbon electrodes	Amperometry	24 nM	0.1–80 µM	[29]
Carbon nanotube-based electrode	Amperometry	0.06 µM	0.1–60 µM	[26]
Carbon nanotube-based electrode	Amperometry	4.6 µM	5.0–200 µM	[27]
Multiwall carbon nanotube paste electrode	Voltammetry	0.8 µM	0.1–210 µM	[28]
Graphene nanosheet-supported platinum nanoparticle electrode	Voltammetry	0.2 nM	0.2–2.4 nM	[30]
CuO/ZnO nanocomposite	Optical	40 µM	40–96 µM	[31]
**Aptamer-Based Biosensor Platform**				
Aptamer-modified Au NP/graphene sponge electrode	Voltammetry	1.0 µM	1–100 µM	[39]
Aptamer-modified gold nanoparticle/carbon electrode	Voltammetry	0.009 µM	0.05–20 µM	[40]
Aptamer-modified gold electrode	Voltammetry	10 nM	0.2–10 µM	[41]
Aptamer–gold nanoparticle	Optical	0.3 µM	0.5–3.0 µM	[42]
**Quantum Dot Platform**				
Nitrogen-doped graphene quantum dots	Optical	0.05 nM	0.05–50 nM	[15]
Cysteamine-stabilized CdTe quantum dots	Optical	3.3 nM	6.7–400 nM	[32]
Graphene quantum dots	Optical	5 nM	0–50 nM	[33]
**Molecularly Imprinted Polymer-Based Biosensor Platform**				
MIP-based optical sensor	Optical	NA	NA	[45]
MIP-modified nanocomposite screen-printed carbon electrode	Voltammetry	1.2 µM	5.0–150 µM	This Work

**Table 2 polymers-15-02241-t002:** The accuracy of the Hcy-MIP biosensor for homocysteine detection at concentration of 50, 75, 100, and 150 µM.

Homocysteine Added (µM)	Homocysteine Obtained (µM)	Recovery (%)
50	46.10	91.10
75	72.60	96.07
100	94.40	93.85
150	144.30	95.83
Average % recovery (*n* = 8)		94.21

Spiked serum samples at Hcy concentrations of 50, 75, 100, and 150 µM were assayed for Hcy by the Hcy-MIP biosensor using the previously described method. The % recovery was calculated as an indicator of accuracy of the assay method (*n* = 8).

**Table 3 polymers-15-02241-t003:** The analytical precision of the Hcy-MIP biosensor for homocysteine detection at concentration of 5 and 150 µM.

Homocysteine Concentration (µM)	Intra-Assay (*n* = 20)		Inter-Assay (*n* = 20)	
	Mean ± SD (µM)	%CV	Mean ± SD (µM)	%CV
5	4.97 ± 0.11	2.27	4.98 ± 0.17	3.42
150	150.35 ± 5.26	3.50	150.40 ± 6.35	4.22

The standard Hcy solution at concentration of 5 and 150 µM was determined for both intra- and inter-assay with 20 measurements using the assay method previously described.

## Data Availability

The data presented in this study are available from the corresponding author upon reasonable request.

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
