# Peer review of "A Label-Free Electrochemical Biosensor for Homocysteine Detection Using Molecularly Imprinted Polymer and Nanocomposite-Modified Electrodes"

_polymers, 2023, doi:10.3390/polym15102241_

Round 1
Reviewer 1 Report
A novel Hcy specific MIP (Hcy-MIP) was synthesized using methacrylic acid (MAA) as functional monomer in the presence of trimethylolpropane trimethacrylate (TRIM) as a cross linker. The Hcy-MIP biosensor was fabricated by overlaying the mixture of Hcy-MIP and carbon nanotube/chitosan/ionic liquid compound (CNT/CS/IL) nanocomposite on the surface of screen-printed carbon electrode (SPCE). However, some critical issues remain to be solved and a thorough revision was needed:
1. The “introduction” and “results and discussion” sections of the manuscript can be strengthened and supported with some papers related to the literature and cited (optional for authors): Appl. Organomet. Chem. 37(4) (2023) e7029; https://doi.org/10.1002/aoc.7102.
2. The advantages and disadvantages of the synthesized material should be investigated.
3. The manuscript needs thorough revision to improve the text quality and readability of the work.
A novel Hcy specific MIP (Hcy-MIP) was synthesized using methacrylic acid (MAA) as functional monomer in the presence of trimethylolpropane trimethacrylate (TRIM) as a cross linker. The Hcy-MIP biosensor was fabricated by overlaying the mixture of Hcy-MIP and carbon nanotube/chitosan/ionic liquid compound (CNT/CS/IL) nanocomposite on the surface of screen-printed carbon electrode (SPCE). However, some critical issues remain to be solved and a thorough revision was needed:
1. The “introduction” and “results and discussion” sections of the manuscript can be strengthened and supported with some papers related to the literature and cited (optional for authors): Appl. Organomet. Chem. 37(4) (2023) e7029; https://doi.org/10.1002/aoc.7102.
2. The advantages and disadvantages of the synthesized material should be investigated.
3. The manuscript needs thorough revision to improve the text quality and readability of the work.
Author Response
Comments from Reviewer 1:
Some critical issues remain to be solved and a thorough revision was needed:
- The “introduction” and “results and discussion” sections of the manuscript can be strengthened and supported with some papers related to the literature and cited (optional for authors): Appl. Organomet. Chem. 37(4) (2023) e7029; https://doi.org/10.1002/aoc.7102.
Response: Thank you for the suggestion to improve the manuscript by strengthened and supported with some papers related to this work. We note that you suggest the inclusion of a link to https://doi.org/10.1002/aoc.7102 can be include in the results and discussion part of Page 11 line 364-367. The relevant references and citations were added in the revised manuscript.
- The advantages and disadvantages of the synthesized material should be investigated.
Response: The advantages and disadvantages of Hcy-MIP are likely to be the same as general MIP. MIP itself function as artificial antibody to the target analytes. The major advantages are the robustness and tolerance to ambient temperature. The long shelf-life can be expected. The disadvantages are less association constant and specificity than native binding molecule of antibody.
- The manuscript needs thorough revision to improve the text quality and readability of the work.
Response: Thank you for the suggestion of the language editing requirement to improve the quality of manuscript. This revised manuscript was extensively reviewed and edited by our native English-speaking colleague; Mr. Graham K, Rogers (The Foreign Expert).
Reviewer 2 Report
Here are a few questions.
1. How are the long term stability and reliability?
2. After DPV, how was the electrode recovered without reversing the electro oxidation?
3. The SEM images look questionable quality. How were the average diameters calculated? The authors claimed one is 100 nm and the other is 150nm. I couldn't see this 50% difference clearly.
4. What Ag/AgCl reference electrode were used? Concentration of KCl in use? Or solid status? It needs clarified.
5 What do the shoulder peaks at about 0 indicate in figure 8?
6. Capital X in fig 9 is a typo?
Author Response
Comments from Reviewer 2:
Here are a few questions.
- How are the long-term stability and reliability?
Response: to be honest, we did not study the long-term stability and reliability of our Hcy-biosensor. The result in this paper only presented the reliability of the system after finishing the preparation of the Hcy-biosensor which was acceptable for the clinical diagnosis. Regarding to the unique characteristics in robustness and tolerance to ambient temperature. Therefore, the long-term performance can be expected from this biosensor system.
- After DPV, how was the electrode recovered without reversing the electro oxidation?
Response: This study focusses on the disposable electrode for single used only. So, we did not investigate the electrode recovered without reversing the electro oxidation.
- The SEM images look questionable quality. How were the average diameters calculated? The authors claimed one is 100 nm and the other is 150 nm. I couldn't see this 50% difference clearly.
Response: Thank you for pointing out the quality of SEM images. Each SEM image was randomly selected 5 areas to measure the diameter size of particles. The average diameter value of polymer was calculated from 5 repetitive scanning images. So, the total of 25 areas was used for calculation the average diameter. To improve the quakity of this work, the Fig 2 of Page 6 was changed for the better SEM image. It clearly presents the different in size between MIP and NIP.
- What Ag/AgCl reference electrode were used? Concentration of KCl in use? Or solid status? It needs clarified.
Response: Thank you for the question. In this work, the Ag/AgCl reference electrode is solid state from the screen-printed electrode. Quasense supplied the screen-printed electrode. (ref: http://www.quasense.co.th/product/screen-printed-electrode/)
Triple Electrode System: Screen Printed Carbon Electrode
Product’s code: CI1703OR
Overall dimension of individual SPE: 12.5×30 mm <W×L>
Working electrode material: Carbon
Working electrode size: Ø = 3 mm, Area = 7.065 mm2
Reference electrode material: Silver/Silver Chloride <Ag/AgCl>
Counter electrode material: Carbon
Electrode connection material: Silver <Ag>
This additional information was added into the revised manuscript at Page 3 line 101 – 102.
- What do the shoulder peaks at about 0 indicate in figure 8?
Response: Thank you for bringing this to our attention. The small peak at 0.0 V could be caused by nanocomposites containing liquid ion, however this is currently being investigated. We focused on the 0.3 V peak in this study since it demonstrates the concentration dependence of the Hcy sample.
- Capital X in fig 9 is a typo?
Response: The correction of typing error in Fig 9 Page 11 was done.
Reviewer 3 Report
The authors developed a novel label free electrode assay able to selectively detect Hcy. The work is meaningful and might find applications in the industry. Since the work is well performed there are only a few small points which require mending as noted below:
1. The abstract is too much in details, that the importance of the research (why it was conducted) is missing at the beginning.
2. First sentence lacks reference.
3. Page 1 line 36 reference missing.
4. Page 2 line 56 reference missing.
5. Page 2 line 60 reference missing.
6. Page 2 line 64 reference missing.
7. Sigma-Aldrich was bought by Merck, please correct this. Moreover, used chemicals should mentioned correctly: (purity/molecular weight for polymers, company, city, country)
8. Line 91: The unit is not correctly formatted, since the multiplication sign is missing.
9. The authors should write in methods and materials if they worked in reflection or transmission mode for FTIR.
10. The used devices should mentioned correctly (also the not mentioned ones should be mentioned): (type, company, city, country)
11. Page 6 line 233 and 234 and 235 and 236, formatting the unit correctly.
12. I doubt the pH mentioned in line 93 for the PBS solution. The pH should be in the range of 7.2 – 7.4?
13. Page 6 line 238 point forgotten.
14. Page 10 line 323 format superscription correctly.
15. Figure 1 looks a bit cutted at the upper edge.
16. Figure 2: The scale bar is too tiny and not readable. Please correct this. In addition, the images are not very contrasting and unsharp. Please correct this with images of better quality.
17. Figure 3: Format superscription correctly for the unit in x.
18. The unit liter should correctly written with an uppercase L throughout the whole manuscript.
19. The authors could state if other sensors like polymeric photonic crystal systems1 or surface sensitive rotators 2 could be coated by the presented approach and respond with similar signals.
References
(1) Zhang, J.; Gai, M.; Ignatov, A. V; Dyakov, S. A.; Wang, J.; Gippius, N. A.; Frueh, J.; Sukhorukov, G. B. Stimuli-Responsive Microarray Films for Real-Time Sensing of Surrounding Media, Temperature, and Solution Properties via Diffraction Patterns. ACS Appl. Mater. Interfaces 2020, 12 (16), 19080–19091. https://doi.org/10.1021/acsami.0c05349.
(2) Ahn, J.; Xu, Z.; Bang, J.; Ju, P.; Gao, X.; Li, T. Ultrasensitive Torque Detection with an Optically Levitated Nanorotor. Nat. Nanotechnol. 2020, 15 (2), 89–93. https://doi.org/10.1038/s41565-019-0605-9.
I just found some minor issues, like missing space signs (mostly in the tables).
Author Response
Comments from Reviewer 3:
Since the work is well performed there are only a few small points which require mending as noted below:
Response: Thank you very much for the constructive remarks and useful suggestions, which have significantly raised the quality of the manuscript. Each suggested revision and comment were accurately incorporated and considered, and some changes according to the comments have been performed in the manuscript.
- The abstract is too much in details, that the importance of the research (why it was conducted) is missing at the beginning.
Response: The abstract of Page 1 was edited by cutting-out some details information and adding the rational of this work.
- First sentence lacks reference.
Response: The reference of first sentence was stated after the second sentence at line 30.
- Page 1 line 36 reference missing.
Response: The reference of the sentence in Page 1 line 36 was stated in line 37.
- Page 2 line 56 reference missing.
Response: The reference of the sentence in Page 2 line 57 was stated in line 57.
- Page 2 line 60 reference missing.
Response: The reference of the sentence in Page 2 line 60 is the same as in line 57. The references were added.
- Page 2 line 64 reference missing.
Response: The reference of the sentence in Page 2 line 64 was stated in line 64.
- Sigma-Aldrich was bought by Merck, please correct this. Moreover, used chemicals should mentioned correctly: (purity/molecular weight for polymers, company, city, country)
Response: Thank you for latest information. The information of purity/molecular weight for chemical, company, city, country) were added in section 2.1 Chemicals and specimens, Page 2, line 81-88 of revised manuscript. Actually, the chemicals in our inventory during the time of experiment are from Sigma-Aldrich. To update this information, the chemicals from Sigma-Aldrich were changed to Merck.
- Line 91: The unit is not correctly formatted, since the multiplication sign is missing.
Response: Page 2 line 88-91 is the phosphate buffer solution. We usually used the sentence of “………… were dissolved in 0.1 M phosphate buffer solution (PBS), pH 7”. It is not clear to us where is the multiplication sign missing?
- The authors should write in methods and materials if they worked in reflection or transmission mode for FTIR.
Response: We did the FTIR in Attenuated Total Reflectance (ATR) mode. This information was added in the methods and materials section at Page 4 section 2.3.3 line 154–156.
- The used devices should mentioned correctly (also the not mentioned ones should be mentioned): (type, company, city, country)
Response: The suggestion was done in Page 2 line 96-98 and Page 3 line 99–103.
- Page 6 line 233 and 234 and 235 and 236, formatting the unit correctly.
Response: The unit of FTIR spectra are correct.
- I doubt the pH mentioned in line 93 for the PBS solution. The pH should be in the range of 7.2 – 7.4?
Response: Thank you for pointing out the pH for PBS solution. The solution in line 89 is pH 7.0.
- Page 6 line 238 point forgotten.
Response: Please specified point forgotten in Page 6 line 238.
- Page 10 line 323 format superscription correctly.
Response: Done
- Figure 1 looks a bit cutted at the upper edge.
Response: The Figure 1 was changed with the perfect one.
- Figure 2: The scale bar is too tiny and not readable. Please correct this. In addition, the images are not very contrasting and unsharp. Please correct this with images of better quality.
Response: The Fig 2 was changed for the better SEM image. It clearly presents the different in size between MIP and NIP.
- Figure 3: Format superscription correctly for the unit in x.
Response: The Figure 3 was corrected with the superscript of the unit in x-axis.
- The unit liter should correctly type with an uppercase L throughout the whole manuscript.
Response: All of the unit liter was changed to an uppercase L.
- The authors could state if other sensors like polymeric photonic crystal systems1or surface sensitive rotators 2 could be coated by the presented approach and respond with similar signals.
References
(1) Zhang, J.; Gai, M.; Ignatov, A. V; Dyakov, S. A.; Wang, J.; Gippius, N. A.; Frueh, J.; Sukhorukov, G. B. Stimuli-Responsive Microarray Films for Real-Time Sensing of Surrounding Media, Temperature, and Solution Properties via Diffraction Patterns. ACS Appl. Mater. Interfaces 2020, 12 (16), 19080–19091. https://doi.org/10.1021/acsami.0c05349.
(2) Ahn, J.; Xu, Z.; Bang, J.; Ju, P.; Gao, X.; Li, T. Ultrasensitive Torque Detection with an Optically Levitated Nanorotor. Nat. Nanotechnol. 2020, 15 (2), 89–93. https://doi.org/10.1038/s41565-019-0605-9.
Response: Thank you for the value suggestion. We stated the possible to coat the presented on polymeric photonic crystal systems in the results and discussion part of Page 11. The relevant references and citations were added in the revised manuscript.
Comments on the Quality of English Language
I just found some minor issues, like missing space signs (mostly in the tables)
Response: This revised manuscript was extensively reviewed and edited by our native English-speaking colleague; Mr. Graham K, Rogers (The Foreign Expert). The missing space signs in the table 1 (Page 12) and the table 3 (Page 13) were corrected.
Round 2
Reviewer 2 Report
The response is satisfying